# Efficient Repairing of Disconnected Pulmonary Tree Structures via Point-based Implicit Fields

**Ziqiao Weng**[1]                                           ZWEN6522@UNI.SYDNEY.EDU.AU
**Jiancheng Yang**[*2]                                       JIANCHENG.YANG@EPFL.CH
**Dongnan Liu**[1]                                           DONGNAN.LIU@SYDNEY.EDU.AU
**Weidong Cai**[1]                                           TOM.CAI@SYDNEY.EDU.AU

[1] *University of Sydney, Australia*

[2] *Swiss Federal Institute of Technology Lausanne (EPFL), Switzerland*

## Abstract

Segmentation of pulmonary tree structures is critical for diagnosing and planning treatment for lung diseases. However, existing deep learning models often yield inaccurate segmentations, resulting in disconnected vessel predictions. To overcome this challenge, we propose an efficient framework for reconstructing pulmonary trees. Initially, we represent disconnected pulmonary tree structures as sparse surface point clouds. Next, we utilize a point cloud network to extract features and predict the disconnected segments. Finally, we employ an implicit neural network to infer the occupancy of arbitrary points, thereby facilitating efficient reconstruction. We validate the effectiveness of our approach on real data from 799 subjects; the code and data will be publicly available.

**Keywords:** Pulmonary Tree Segmentation, Pulmonary Tree Reconstruction, Lung Diseases, Point Cloud, Implicit Neural Network.

## 1. Introduction

Automated segmentation of pulmonary tree structures, including airways and vessels, is essential for diagnosing and treating pulmonary diseases (Weng et al., 2023; Xie et al., 2023). In recent years, numerous deep learning-based methods have focused on segmenting pulmonary vessels from lung CT images (Qin et al., 2021; Cui et al., 2019). However, due to the complexity and sparsity of pulmonary tree structures, along with the low contrast of CT images, even advanced segmentation models like nn-UNet (Isensee et al., 2021) struggle to accurately segment pulmonary vessels, resulting in disconnected vessel predictions.

To tackle this challenge, (Weng et al., 2023) addressed disconnected pulmonary tubular structures as a key point detection task and achieved initial success. However, the work did not address repairing disconnected segments, posing ongoing challenges for clinicians interpreting vessel segmentation results. Additionally, their method's efficiency is compromised by the high resolution and sparsity of 3D volume data, leading to significant computational overhead and constraining input data size (with the study confined to cropping a smaller volume from the original).

In this paper, we overcome these challenges and efficiently reconstruct the pulmonary tree. Firstly, we convert volume data into point cloud data and extract down-sampled surface points to achieve full-resolution input. Secondly, we utilize keypoint detection for

---

* Corresponding author

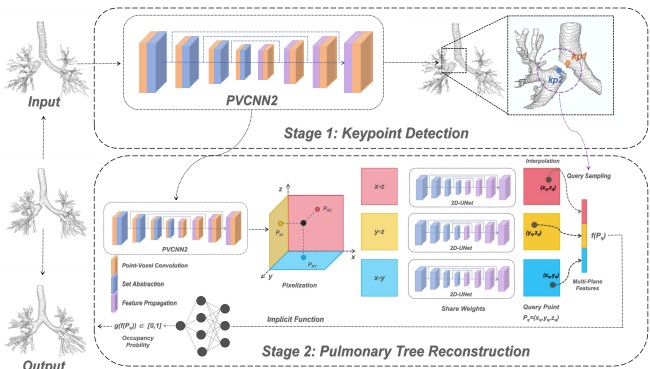

Figure 1: Pulmonary Tree Reconstruction (PTR) framework.

pre-training to extract features for the reconstruction network. Thirdly, we obtain multi-plane features of the input point cloud. Finally, in disconnected vessel regions, we use an implicit function to infer new points and facilitate vessel reconstructions.

## 2. Method

**Data processing**: We create our pulmonary point cloud dataset based on the PTR dataset introduced by (Weng et al., 2023). We extract surface points from the 3D volume and downscale them by a factor of three as the point cloud coordinates of input for network. we also introduce a new "super point" method to address information loss during downsampling where we record the occupancy of neighboring points (with R=3 in our experiments) around each point as the point cloud features. Consequently, the coordinates and features of point cloud are served as input to the following network.

**Network strcuture**: Figure 1 provides an overview of our Pulmonary Tree Reconstruction (PTR) framework. Initially, we conduct keypoint detection, defining keypoints as endpoints along the interrupted vessel centerline. Employing PVCNN (Liu et al., 2019) as our point cloud network, we predict both heatmap values and offsets for each point simultaneously. The final predicted keypoint is determined by combining the point with the highest heatmap value and its corresponding offset. MSE Loss is used for heatmap value regression, while L1 Loss is applied for offset regression. In the second stage, the PVCNN, loaded with weights trained in the previous stage, extracts features for each point. Following (Peng et al., 2020), we project point cloud features onto three planes and perform separate pixelization through average pooling. The features from each plane are then processed using a 2D-UNet with shared weights. We extract query points from around disconnected vessel branches, interpolate features from the three planes, and concatenate them as input to the implicit function represented by an MLP network. Ultimately, the network predicts the occupancy of each point, where 1 indicates a vessel point and vice versa. We utilize a combination of Dice Loss and BCE Loss for network training. During the inference stage, query points are sampled from the predicted disconnected regions.

## 3. Experiment

**Dataset**: Our point cloud PTR dataset comprises 799 manually annotated binary pulmonary trees extracted from CT images. Each instance consists of 30 disconnected branches,

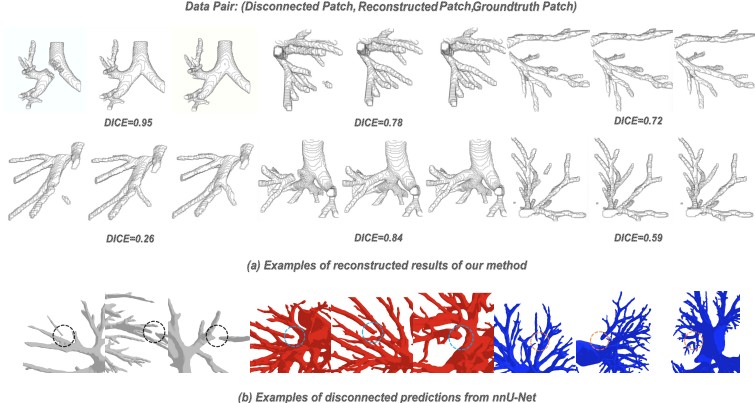

Figure 2: (a) Examples of reconstructed results of our method. Dice refers to $Dice_{local}$; (b) Examples of disconnected predictions from nnU-Net.

resulting in a total of 23,970 data points. The dataset is divided into three subsets for training, validation, and testing, with a subject-wise ratio of 7:1:2.

**Keypoint detection**: We adhere to the metrics specified in (Weng et al., 2023). Initially, the experiment is carried out at the cropped subvolume level of $80^3$ (referred to as 'Local' in Table 1). Subsequently, we assess the performance of PVCNN in the entire pulmonary tree space, denoted as 'Global' in Table 1. The experimental results compellingly demonstrate the effectiveness and efficiency of our approach.

**Reconstruction**: We assess the reconstruction quality using several metrics, including the F1 score (calculated on query points), Chamfer Distance (CD), IOU, Dice for the entire pulmonary tree ($IOU$, $Dice_{global}$), and Dice for disconnected branch regions ($Dice_{local}$), as depicted in Table 2. The term 'None' indicates the result calculated between disconnected and complete pulmonary trees. Furthermore, in Figure 2, we provide visualizations of the reconstruction results and examples of disconnected predictions from the nn-UNet model.

Table 1: Keypoint detection performance on the PTR dataset.

| Level | Method | $AP(\%)\uparrow$ | $AP^{50}$ | $AP^{75}$ | $E_d(\%)\uparrow$ | $GPU_{memory}\downarrow$ | $Params\downarrow$ | $MAC\downarrow$ |
|---|---|---|---|---|---|---|---|---|
| Local | UNet | 87.18 | 98.48 | 94.89 | 28.54 | 23.99 G | 19.07 M | 580.96 G |
| | PVCNN | 89.24 | 97.49 | 94.56 | 38.86 | 4.84 G | 13.70 M | 26.24 G |
| Global | PVCNN | 87.70 | 94.15 | 92.08 | 35.57 | 14.54 G | 13.73 M | 173.88 G |

Table 2: Reconstruction performance on the PTR dataset.

| Method | $F1(\%)$ | $CD\downarrow$ | $IOU(\%)\uparrow$ | $Dice_{global}(\%)\uparrow$ | $Dice_{local}(\%)$ |
|---|---|---|---|---|---|
| None | \ | 4.57 | 99.70 | 99.84 | \ |
| PTR-Net | 86.60 | 2.03 | 99.91 | 99.95 | 75.86 |

## 4. Conclusion

We have presented a two-stage framework for reconstructing disconnected pulmonary tree segmentations. By leveraging sparse point clouds, diverse multi-plane features, and efficient implicit neural networks, we effectively address disconnected segments. Our ongoing efforts involve reconstructing multiple disconnected pulmonary trees and exploring further meaningful tasks based on this foundation.

## Acknowledgments

This research was supported by Australian Government Research Training Program (RTP) scholarship, and supported in part by a Swiss National Science Foundation grant.

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
