# OpenReview forum: "Efficient Repairing of Disconnected Pulmonary Tree Structures via Point-based Implicit Fields"
_MIDL.io/2024/Short_Papers — MIDL 2024 Short Papers_

### Official Review · Reviewer_DBwJ · 2024-04-22

**Confidence:** 4
**Final Rating:** 4

**Review:**

Summary:

This paper introduces a novel method for utilizing a point cloud network to extract features and predict disconnected segments. It presents an efficient approach to repairing disconnected parts of the pulmonary airway with reduced computational resources.

Pros: (1) Using a point cloud network for segmenting disconnected pulmonary airways is an innovative approach that effectively combines keypoint detection and reconstruction to achieve optimal results.

Cons: (1) In Figure 1, the author needs to explain the meanings of the yellow, blue, and purple modules in the PVCNN module. Each colour represents a different component of the network. (2) In Figure 2, it is recommended to modify the Dice scores to compare the values before and after repair. (3) Since the text mentions predicted heatmap values, these should be displayed in the results section, either through figures or tables, to substantiate the claims made. If it's not possible to include this data, then the mention of heatmap values should be removed from the text to maintain consistency and clarity in the presentation of the research findings. (4) As far as I know, there are also other methods to repair disconnected branches in 3D pulmonary airways. Could you compare the method proposed in this article with the state-of-the-art methods?

---

### Decision · Program_Chairs · 2024-04-26

Accept